# GAWMerge expands GWAS sample size and diversity by combining array-based genotyping and whole-genome sequencing

Ravi Mathur [1,10], Fang Fang [1,10], Nathan Gaddis[1], Dana B. Hancock [1], Michael H. Cho [2,3], John E. Hokanson[4], Laura J. Bierut[5], Sharon M. Lutz[6], Kendra Young [4], Albert V. Smith[7,8], NHLBI Trans-Omics for Precision Medicine (TOPMed) Consortium*, Edwin K. Silverman[2,3], Grier P. Page [1,9] & Eric O. Johnson [1,9 ✉]

Genome-wide association studies (GWAS) have made impactful discoveries for complex diseases, often by amassing very large sample sizes. Yet, GWAS of many diseases remain underpowered, especially for non-European ancestries. One cost-effective approach to increase sample size is to combine existing cohorts, which may have limited sample size or be case-only, with public controls, but this approach is limited by the need for a large overlap in variants across genotyping arrays and the scarcity of non-European controls. We developed and validated a protocol, Genotyping Array-WGS Merge (GAWMerge), for combining genotypes from arrays and whole-genome sequencing, ensuring complete variant overlap, and allowing for diverse samples like Trans-Omics for Precision Medicine to be used. Our protocol involves phasing, imputation, and filtering. We illustrated its ability to control technology driven artifacts and type-I error, as well as recover known disease-associated signals across technologies, independent datasets, and ancestries in smoking-related cohorts. GAWMerge enables genetic studies to leverage existing cohorts to validly increase sample size and enhance discovery for understudied traits and ancestries.

[1] GenOmics, Bioinformatics, and Translational Research Center, RTI International, Research Triangle Park, NC, USA. [2] Channing Division of Network Medicine, Brigham and Women's Hospital, Boston, MA, USA. [3] Division of Pulmonary and Critical Care Medicine, Brigham and Women's Hospital, Boston, MA, USA. [4] Department of Epidemiology, Colorado School of Public Health, University of Colorado Denver, Aurora, CO, USA. [5] Department of Psychiatry, Washington University, St. Louis, MO, USA. [6] PRecisiOn Medicine Translational Research (PROMoTeR) Center, Department of Population Medicine, Harvard Medical School and Harvard Pilgrim Health Care, Boston, MA, USA. [7] Center for Statistical Genetics, University of Michigan, Ann Arbor, MI, USA. [8] Department of Biostatistics, University of Michigan, Ann Arbor, MI, USA. [9] Fellow Program, RTI International, Research Triangle Park, NC, USA. [10] These authors contributed equally: Ravi Mathur, Fang Fang. *A list of authors and their affiliations appears at the end of the paper. ✉email: ejohnson@rti.org

Genome-wide association studies (GWAS) offer a powerful tool for identifying genetic variants for complex diseases, especially when large sample sizes are amassed. For diseases with limited sample sizes or for which case-only cohorts are available, public controls, who are not assessed for the disease, can be used without bias to cost effectively improve statistical power and novel locus discovery, if the disease prevalence is low in the general population[1–5]. Case-only GWAS datasets may exist for several reasons but primary among them is that the initial study focused on phenotypes within a patient population (e.g., set point viral load among those living with HIV[6,7] or methadone dosing among those with opioid use disorder (OUD)[8,9] but these case-only datasets could now be useful for GWAS of the primary disease (HIV or OUD) if paired with public controls. Combining cases and controls in this way is feasible even with samples genotyped on different array-based technologies[10–13]. A significant limitation of combining disease study cases with public controls is that unbiased results are only achieved using the intersecting set of variants genotyped across all arrays and cohorts being combined[13]. This limitation effectively prevents combining cohorts where the number of shared genotyped variants is too small to form the basis for imputation or to provide whole-genome coverage. An in-depth comparison of the Illumina HumanHap, Illumina OmniExpress, and Affymetrix 6.0 arrays found over 2,000,000 single nucleotide polymorphisms (SNPs) in union but only 75,000 variants that intersect across all arrays[14]. Additionally, reliance on array-based technology prevents use of expanding whole-genome sequencing (WGS) resources with high representation of non-European ancestry groups, like the Trans-Omics for Precision Medicine (TOPMed) program, for public controls. Being able to combine case and public control genotypes from array- and/or sequencing-based platforms opens up the increasing set of WGS resources for new GWAS. As of January 2021, there are at least 217 case-only studies containing >136,000 samples across many genotyping platforms in the database of Genetics and Phenotypes (dbGaP) (query = 'case set[Study Design]'). As of February 2020, freeze 6a, there are >227,000 public controls with WGS data in resources such as TOPMed (>155,000 samples)[15], UK BioBank (>50,000 samples)[16], Gabriella Miller Kids First Pediatric Consortium (>21,000 samples), and GenomeAsia 100K Project (>1,700 individuals)[17], which are eligible to be combined with these case-only datasets for GWAS. Both the pools of case-only data and WGS public controls will continue to increase. Although the array genotyping may have a lower overall precision due to the poor cluster separation in the genotype assignment pipeline based on a 2-dimensional metrics, the average discordant calls were below 1%[18], which supports the feasibility to combine the array genotyping data with WGS data.

The NHLBI-supported TOPMed program[15] with its collection of >155,000 human subjects with WGS data affords an unparalleled opportunity to leverage public controls and greatly expand GWAS sample sizes. With such a large sample size and one of the most genetically diverse datasets (40% European, 31% African, 16% Hispanic, 9% Asian, and 4% Others) available, TOPMed has the potential to overcome the aforementioned challenges of applying public controls, as the WGS data should overlap all variants measured on arrays, and the representation of non-European populations will enhance the availability of diverse public controls.

While incorporating public controls to maximize the utility of genetic discovery is desirable, there is no established approach to validly combine array- and sequencing-based genotype data. Each of these technologies has its own strengths, weaknesses, and different inter- and intra-technology measurement properties that complicate combining data across technologies. Here, we developed a protocol, Genotype Array-WGS Merge (GAWMerge), to combine genotypes from array and WGS to conduct GWAS analyses. We illustrate our protocol's validity and its utility using TOPMed WGS samples as public controls combined with case-only array-genotyped data for GWAS of the Chronic Obstructive Pulmonary Disease (COPD) phenotype. COPD has well-established GWAS hits, therefore easily testing replication of signal, and it has high sample size for both European-ancestry and African-ancestry groups within the TOPMed program.

## Results

**Protocol to integrate array and WGS data**. GAWMerge is a protocol that we developed to integrate array and WGS genotyping technologies that minimizes false positives while discovering true association signals. Details of the protocol development process are provided in the Methods section. The final protocol consists of eight major steps (Fig. 1): (1) select control dataset(s) with WGS genotype data; (2) extract the SNPs from the WGS data of the control samples that match those for the array-genotyped case samples; (3) independently subject the case and control samples to the same quality control (QC) procedure (further details in the Methods); (4) phase the case and control samples with the same software (further details in the Methods); (5) merge the phased case and control data and impute to the desired reference genome (e.g., 1000Genome, TOPMed reference panel); (6) filter out genotyped SNPs with low quality (empirical $ER^2 < 0.9$)[19] and re-impute; (7) test SNP associations with phenotype of interest in case and control samples combined; and (8) filter association results for minor allele frequency (MAF), imputation quality ($R^2$), and difference in imputation quality.

For selection of controls in step 1, it is crucial to choose samples with an ancestral composition consistent with the case samples, as population stratification is a strong confounding factor for GWAS analysis. Additional demographic (e.g., age, sex) and clinical variables (e.g., smoking status) should be considered based on the datasets being combined.

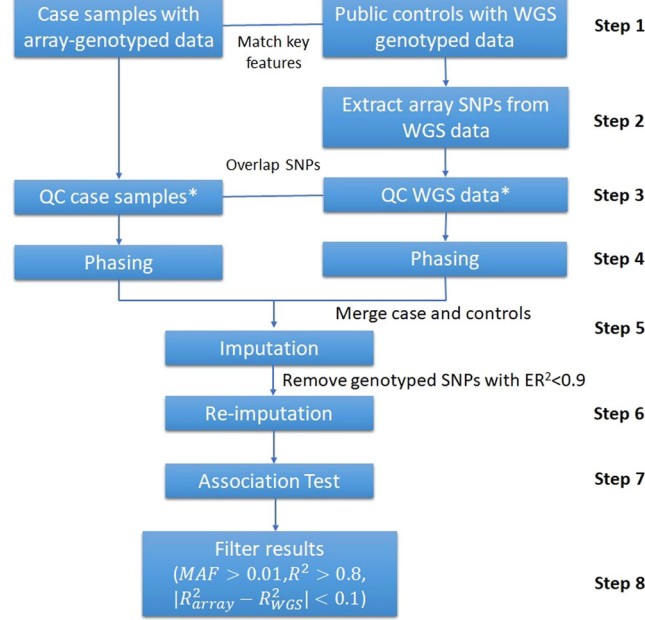

**Fig. 1 Overview of the protocol to use whole-genome sequencing (WGS) data as public control in GWAS.** *The quality control (QC) of the case and public control data is conducted independently according to the steps outlined in the methods.

Our previous work[13] suggested potential bias in association testing when using genotypes imputed from the full sets of SNPs from different genotyping arrays. Starting from the intersection of genotyped SNP sets avoids such bias (step 2). We employed the same strategy for merging array and WGS genotypes, but because of the full genome coverage of WGS, the entire set of array SNPs were used. We also investigated the intersection between the WGS data in TOPMed and different targeted arrays (e.g., MetaboChip, Immnochip and OncoArray), and the overlapping rates were all above 95% (Supplementary Table 1), therefore validating the integrating of array and WGS data. The array and WGS data were then independently QC'd using the same QC steps (step 3). This then was followed by phasing, merging, and imputation (steps 4–5). To further reduce potential bias between the array-genotyped and WGS-derived SNPs, a second round of imputation is performed after removing genotyped SNPs with low empirical $R^2$ (ER$^2$ < 0.9, step 6, Supplementary Fig. 1). Finally, following association testing (step 7), filtering based on MAF ( > 0.01), imputation quality ($R^2$ > 0.8), and imputation quality difference between cases (i.e., array data) and controls (i.e., WGS data) is step 8 ($|R^2_{array} - R^2_{WGS}| < 0.1$, Supplementary Fig. 2) which minimizes technical variation in the combined case/control data. More details regarding the development of the protocol can be found in the "Protocol Development" section of Methods.

**Protocol evaluation design**. To evaluate the performance of GAWMerge, we used three smoking-related datasets: Collaborative Genetic Study of Nicotine Dependence (COGEND)[20,21], Genetic Epidemiology of COPD (COPDGene) study[22], and Evaluation of COPD Longitudinally to Identify Predictive Surrogate End-points (ECLIPSE)[23]. As indicated in Table 1, the three datasets have different array platforms, providing the opportunity to assess the performance of the protocol in different settings. In both COPDGene and ECLIPSE, the COPD diagnosis followed the Global Initiative for Chronic Obstructive Lung Disease severity classifications, and COPD cases were defined as severity Grade 2–4 COPD (moderate, severe, and very severe COPD)[24]. The study design to evaluate GAWMerge across (a) genotyping technology (ensuring no technology driven false positives), (b) type-I error (ensuring minimal false positive associations), and (c) recovery of known GWAS hits (demonstrating capture of true positives) is presented in Fig. 2.

**Reproducibility across genotyping technologies**. COPDGene has both array and WGS genotype data on the same samples available through TOPMed. Genotypes derived from array and whole-genome sequencing data for the same samples should be consistent but are often not[18,25]. To evaluate the consistency of genotyping, we performed a technical comparison of array and WGS data using the same set of samples from COPDGene ($n$ = 3235 with African-American ancestry). The array data were phased independently and integrated with the WGS phased data available in TOPMed, followed by imputation and association testing using genotyping platform as the outcome. If the array- and WGS-derived genotypes for the same set of samples were equivalent, one would expect to observe no significant associations, but in fact we observed many false positives (Supplementary Fig. 3).

We suspected that the false positives we observed derived from the phasing step since phasing of the array and WGS genotypes was based on different sets of variants. In addition, the TOPMed phased WGS data were derived from the samples of all studies[26], which is different from the sample set we used, the COPDGene cohort, for phasing the array data. We repeated the technical comparison, using the same set of QC-validated variants and samples (Fig. 2a) as the basis for separate phasing of the array and WGS data, followed by the subsequent steps in GAWMerge (Fig. 1). The array data were specified as the case group for association testing, and the WGS data were specified as the control group, for European ancestry (EA) and African-ancestry (AA) separately. The results (Supplementary Fig. 4) confirmed that phasing based on a common set of variants and samples followed by the additional steps of GAWMerge eliminated false positives and made array and WGS data comparable for conducting GWAS.

**Controlling type-I error in case-only vs. public control GWAS**. We assessed type-I error in a comprehensive analysis involving three smoking-related datasets and their meta-analysis, as shown in Fig. 2b. To fully leverage the large sample size of the COPD-Gene dataset, we evenly divided the EA samples into two subsets: EA1 and EA2. COPDGene EA1 included all participants diagnosed with COPD ($N$ = 2736) and randomly sampled participants with no COPD ($N$ = 515). The resulting ratio of individuals with COPD in COPDGene EA1 (84%) was close to the ratio in ECLIPSE EA (87%). Three GWAS were conducted to assess type-I error, as follows: (1) array data from COPDGene EA1 ($N$ = 3251) vs. WGS from ECLIPSE EA ($N$ = 1461); (2) array data from COGEND EA ($N$ = 1961) vs. WGS data from COPDGene EA2 (with no COPD, $N$ = 3251); and (3) array data from COGEND AA ($N$ = 712) vs. WGS from COPDGene AA ($N$ = 1710). All association models include ten principal components as covariates to account for population substructure. COPDGene, COGEND, and ECLIPSE are all smoking cohorts and ratios of COPD were consistent across array and WGS datasets, thus we expected no genome-wide significant association signals (controlled type 1 error). Applying GAWMerge to these data we observed no false positive signals in each separate GWAS analysis (Supplementary Fig. 5) and in their meta-analysis (Fig. 3) results.

**Recovery of known COPD loci in case-only vs. public control GWAS**. The last evaluation step was to recover known GWAS hits for COPD[24,27]. As shown in Fig. 2c, we conducted three GWAS for COPD, as follows: (1) COPD cases from COPDGene EA with WGS data ($N$ = 2736) vs. controls from COGEND EA with array data ($N$ = 1961); (2) COPD cases from ECLIPSE EA with array data ($N$ = 1764) vs. controls from COPDGene EA with WGS data ($N$ = 2475); and (3) COPD cases from COPDGene AA with WGS data ($N$ = 813) vs. controls from COGEND AA with array data ($N$ = 712). Because COPD is highly comorbid with smoking history, only smokers (current and former) were used as controls to compare with COPD cases across these GWAS analyses. All association models include ten principal components as covariates to account for population substructure. Results for each GWAS analysis are presented in Supplementary Fig. 6. Meta-analysis of the 3 analyses successfully recovered 5 out of 7 loci reported as COPD-associated (Fig. 4 and Table 2) at genome-wide significance ($P < 5 \times 10^{-8}$, Supplementary Table 2). The direction of association for all recovered SNPs was the same as previously reported[28]. The two SNPs that did not exceed the genome-wide significance threshold were nominally associated at $P < 0.05$ in our analysis. These two SNPs were missing in Analysis 1 (COPD cases with WGS data from COPDGene EA Vs. smoking controls with array data from COGEND EA) due to the filters applied with the protocol; the reduced power caused by their missingness likely explain the lower significance level observed.

**Table 1 Dataset characteristics.**

| Array type | | | COGEND | COPDGene | ECLIPSE |
|---|---|---|---|---|---|
| | | | Illumina HumanOmni2.5 | Illumina HumanOmni1-Quad_v1-0_B | Illumina HumanHap550v3.0 |
| Array-genotyped data | N, SNPs | | 2,443,179 | 1,051,295 | 561,466 |
| | Participants, total N | | 2,673 | 9,962 | 2,159 |
| | Ancestry group, N (%) | European | 1,961 (73%) | 6,664 (67%) | 2,159 (100%) |
| | | African American | 712 (27%) | 3,298 (33%) | NA |
| | Sex, N (%) | Males | 1,019 (38%) | 5,333 (54%) | 1,367 (63%) |
| | | Females | 1,654 (62%) | 4,629 (46%) | 792 (37%) |
| | COPD diagnosis, N (%) | Yes | NA | 4,280 (43%) | 1,764 (82%) |
| | | No | | 3,632 (36%) | 395 (14%) |
| | Age (mean ± SD) | | 36.6 ± 5.6 | 59.6 ± 9.0 | 62.2 ± 8.2 |
| WGS-genotyped data[a] | Participants, total N | | NA | 9,737 | 1,484 |
| | Ancestry group, N (%) | European | NA | 6,502 (67%) | 1,461 (98%) |
| | | African American | | 3,235 (33%) | 23[b] (2%) |
| | Sex, N (%) | Males | | 5,213 (54%) | 933 (64%) |
| | | Females | | 4,524 (46%) | 528 (36%) |
| | COPD diagnosis, N (%) | Yes | | 4,186 (43%) | 1,271 (87%) |
| | | No | | 3,549 (36%) | 190 (13%) |
| | Age (mean ± SD) | | | 59.6 ± 9.0 | 62.7 ± 7.7 |

[a]All WGS-genotyped data are from TOPMed freeze 6a.
[b]The number of African American in ECLIPSE is too small and excluded from following analysis.

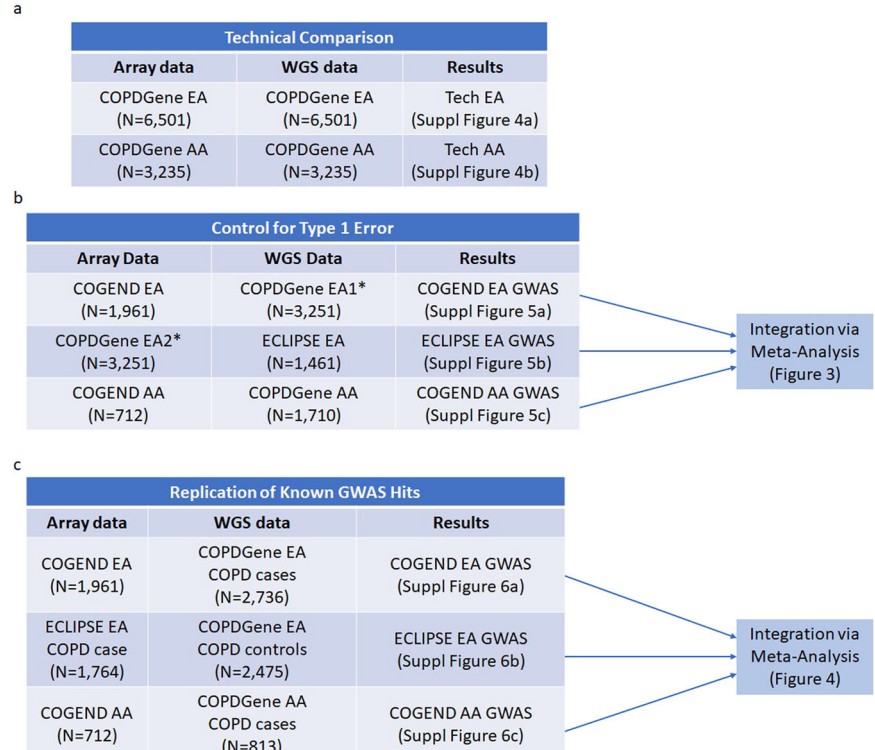

**Fig. 2 Evaluation design for GAWMerge.** Evaluation design for **a** technical comparison, **b** type-I error assessment, and **c** known GWAS hits. *The samples with European ancestry in COPDGene were evenly divided into two subsets of samples. EA1 includes all COPD cases and some COPD controls to match the COPD prevalence in ECLIPSE. EA2 has all the rest COPD free samples.

## Discussion

In summary, we present GAWMerge, a protocol for integrating array and WGS genotype data to conduct GWAS with a case-only and public control design. This protocol overcomes previous obstacles to using public controls[13]. The ability to use WGS data for public controls (1) ensures complete overlap with variants on any array used for genotyping of cases, and (2) provides a much larger pool of public controls to draw from, especially for non-Europeans, from ancestrally diverse resources like TOPMed. In our proof-of-concept study, we applied GAWMerge to WGS data

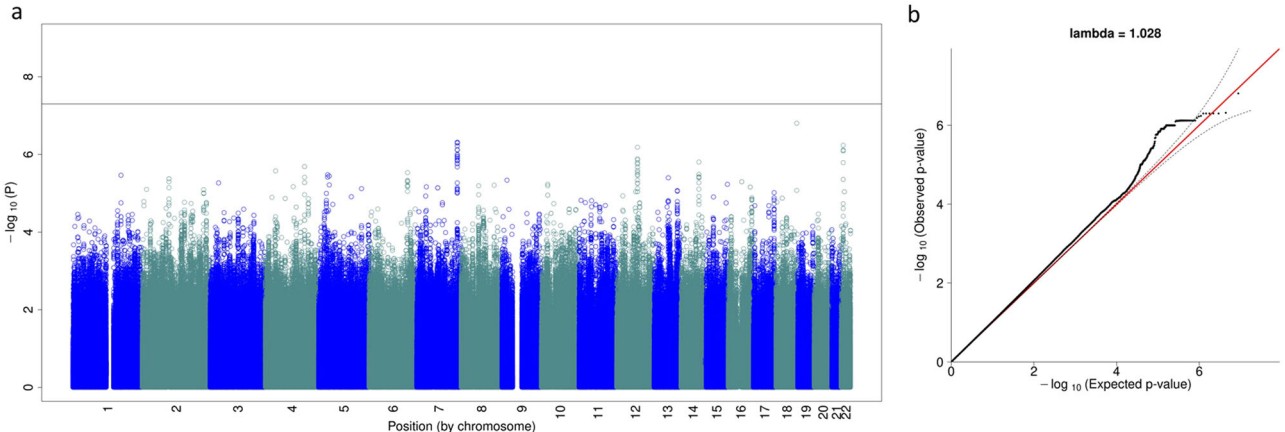

**Fig. 3 Meta-analysis results from evaluation for type-I error.** The Manhattan plot (**a**) shows the expected no signal, while the QQ-plot (**b**) shows no inflation.

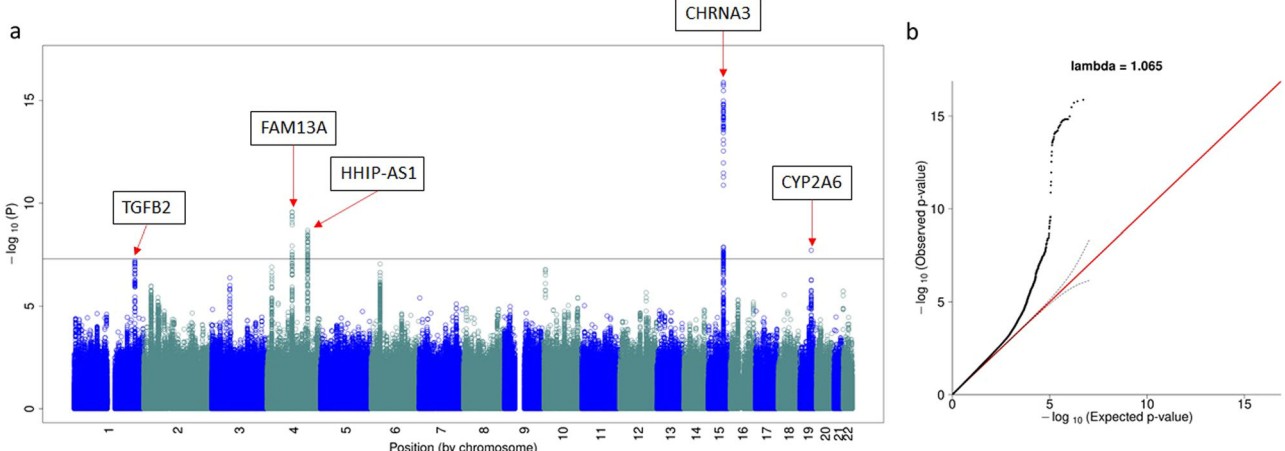

**Fig. 4 Meta-analysis results for replication of GWAS hits for COPD.** The Manhattan plot (**a**) shows the replicated signals, while the QQ-plot (**b**) shows inflation due to the true signal.

**Table 2 Recovery of GWAS-identified variants, following application of our protocol to each of 3 GWAS and their meta-analysis, compared to published risk loci for COPD with combined data from COPDGene, ECLIPSE, NETT/NAS, and GenKOLS (Norway)[24].**

| SNP | Position | Risk Allele | Related gene | Reported ($N = 12,337$) | | Current meta-analysis ($N = 10,461$) | | |
|---|---|---|---|---|---|---|---|---|
| | | | | OR | *P*-value | OR | Direction | *P*-value |
| rs12914385 | chr15:78898723 | T | *CHRNA3* | 1.36 | 2.70E-16 | 1.28 | +++ | 3.35E-16 |
| rs4416442 | chr4:89866713 | C | *FAM13A* | 1.36 | 9.44E-15 | 1.21 | +++ | 2.66E-10 |
| rs7937[27] | chr19:41302706 | C | *CYP2A6* | 0.74 | 2.88E-09 | 0.84 | --- | 1.91E-08 |
| rs4846480 | chr1:218598469 | A | *TGFB2* | 1.26 | 1.25E-07 | 1.19 | +++ | 9.37E-08 |
| rs13141641 | chr4:145506456 | T | *HHIP* | 1.39 | 3.66E-15 | 1.23 | ?++* | 2.64E-07 |
| rs754388 | chr14:93115410 | C | *RIN3* | 1.33 | 6.69E-08 | 1.12 | ?++* | 0.020 |
| rs626750 | chr11:102720945 | G | *MMP3/12* | 1.36 | 5.35E-09 | 1.14 | ?++* | 0.005 |

*The question mark "?" means the SNP is missing from the first analysis, and it may result in reduced power in the final meta-analysis.

from TOPMed (specifically, COPDGene and ECLIPSE cohorts) as public controls for array-genotyped case datasets. We first showed that the two genotyping technologies are compatible by comparing array- and WGS-derived genotypes for the same samples from COPDGene and demonstrating a lack of false positives. We then showed that GAWMerge controls type-I error, as evidenced by the expected lack of genome-wide significant findings in a GWAS meta-analysis comparing smoker cases vs.

smoker controls from independent datasets. Lastly, GAWMerge recovered known COPD-associated findings from Hobbs et al[28]. including *CHRNA3* on chromosome 15, *FAM13A* on chromosome 4, *CYP2A6* on chromosome 19, *TGFB2* on chromosome 1, and *HHIP* on chromosome 4. The key aspects of the protocol that provide these unbiased findings are (1) phasing the array and WGS data independently using only the intersection of variants across technologies and (2) including the empirical $R^2$ and $R^2$ difference filters to remove poorly imputed and differently imputed variants.

The development of GAWMerge was done with TOPMed WGS and array genotyped-data, although it can be applied using any case-only array-genotyped data with other WGS data resources (e.g., UK BioBank[16], Gabrielle Miller Kids First and/or GenomeAsia 100K[17] data). To incorporate new data, it will be important to identify the phenotypic data which will be used to combine controls with available cases. For example, we selected controls based on the smoking status of the cohorts to minimize bias due to smoking. Additional phenotypic and clinical data, such as sex and age distributions, should be considered when selecting the most appropriate controls for combining with available cases. In this study we combined cases and controls with the same ancestry to minimize bias. Further work is needed to evaluate GAWMerge for mega analysis GWAS[29]. GAWMerge was developed with imputation using the thousand genomes reference population, although method can be applied using other reference populations, such as the TOPMed reference population on the Michigan Imputation Server[19]. Since TOPMed samples are used as controls in GAWMerge, there will be sample overlap between the input data and the TOPMed reference population, which may cause bias and must be applied cautiously. Further work is needed to evaluate the bias of such an imputation strategy. The application of GAWMerge can be conducted beyond the phenotypes tested here, where we plan to expand its applications to other studies, such as opioid addiction in the near future.

GAWMerge has some limitations. First, careful consideration of not only ancestry, sex, and age distributions, but other systematic differences between a given case-only cohort and public controls, like smoking status, is essential to unbiased use of public controls and application of GAWMerge. All association analysis conducted included ten principal components as covariates to account for population substructure, although applying GWAS in as homogenous population as possible is desirable. This requirement places some limits on the public controls that can be used for any given case-only cohort. Second, the additional QC steps might mask some real trait-associated variants. In the attempt to recover the known genetic variants associated with COPD, there were two loci (*RIN3* and *MMP3/12*) not reaching the genome-wide significance in the meta-analysis (Table 2). The three SNPs were filtered out in the first GWAS, comparing COPD cases in COPDGene EA with WGS data and smoking controls in COGEND EA with array data, due to high $R^2$ difference between the WGS and array data. Thus, GAWMerge may lose some sensitivity while controlling type-I errors. There is also the potential for reduced power to detect COPD associated genetic variants here due to the missingness of lung function phenotypes in COGEND public controls, with power being reduced relative to the amount of COPD status misclassification among these controls. Third, when GAWMerge has been tested as an application of GWAS, it is limited by the MAF and genomic coverage on array genotyping technologies. Since GAWMerge extracts only SNPs within the array technology, the complete coverage of WGS (over 410 million variants within TOPMed WGS data[26]) is not fully utilized. Therefore, those rare variants and large insertions/deletions only detected in WGS data were lost during the

extraction and merging processes (Supplementary Table 3). However, coming from a case-only dataset with array-based genotyping, the dominant scenario for use of GAWMerge, the WGS is a substantial strength, accounting for all the array genotyped variants except for technology based regional loss of variants. With our strategy of WGS data as public controls for GWAS, there will be regional loss in specific areas depending on the array technology design and quality control of the sequencing. A complete analysis of different regional genetic variants covered specifically by array-genotyping platforms or sequencing will be beneficial to calibrate the application of GAWMerge in the future[30,31].

Overall, GAWMerge presents a practical application of integrating case-only array-genotyped data with WGS data as public controls to enable new GWAS and enhance the potential for discovering novel genetic loci. It is a general approach for integrating array and WGS genotyping technologies. The substantial availability of case-only datasets in public repositories and collected across many consortia makes the protocol broadly applicable. With >155,000 samples with WGS data in the TOPMed program, this is an ample resource for selecting public controls for a variety of case-only disease datasets. With WGS data the overlap of measured variants across genotyping platforms is overcome. Furthermore, the diversity of individuals within the TOPMed (>47,000 African, >23,000 Hispanic/Latino, and >13,000 Asian ancestries) and increasing representation in other resources make widespread use of non-European public controls realistic. With many other WGS resources being launched and released, the potential to use public controls to increase sample size and leverage case-only cohorts is just beginning.

## Methods

**Dataset descriptions**. The Trans-Omics for Precision Medicine (TOPMed) program aims to improve understanding of the diseases through the integration of Whole-Genome Sequencing (WGS) and other omics data from pre-existing parent studies having large samples of human subjects. The two studies used in this work, Genetic Epidemiology of Chronic Obstructive Pulmonary Disease (COPDGene) and Evaluation of COPD Longitudinally to Identify Predictive Surrogate Endpoints (ECLIPSE), are both part of TOPMed. As of February 2020, TOPMed has gathered data from ~155k participants with rich phenotypic data. TOPMed prioritizes to increase ancestral and ethnic diversity, so ~60% of the sequenced participants are of non-European ancestry (31% African, 16% Hispanic, 9% Asian, and 4% Others).

COPDGene (ClinicalTrials.gov: NCT00608764) is an ongoing study of over 10,000 non-Hispanic White and African American cigarette smokers. It was designed to investigate COPD and other smoking-related lung diseases[22]. COPDGene subjects were initially genotyped for ~1 million single nucleotide polymorphisms (SNPs) using the HumanOmniExpress array (Illumina, San Diego, CA). As part of TOPMed freeze 6a, WGS was conducted on 10,372 subjects. Among them, 9,732 subjects overlapped with the subjects in the parent study having array genotyped data, and thus were used in our analyses.

ECLIPSE was an observational study launched in 2006[23]. It recruited 2,164 COPD subjects, 337 smoking controls, and 245 non-smoking controls. The genotype data with Illumina HumanHap550v3.0 array (~550,000 SNPs) included 1,764 COPD subjects, 217 smoking controls, and 178 non-smoking controls. In TOPMed freeze 6a, WGS was conducted on 1,271 COPD subjects and 190 smoking controls.

COGEND was initiated in 2001 as a genetic study of nicotine dependence[20,21]. Nicotine dependent cases and non-dependent smoking controls were identified and recruited from Detroit and St. Louis. Over 2,900 donated blood samples were collected and used to genotype ~2.5 million SNPs using the HumanOmni2.5 array. After QC, 2,673 subjects were kept for following analyses.

The use of the TOPMed WGS data was approved by the TOPMed Methods working group. Data approval of the dbGaP available data was approved by the RTI-International Institutional Review Board. Informed consent for general research use was obtained for all data by the original study.

**GAWMerge development**. Below we provide further details on the protocol steps, and iterations used to devise the recommended thresholds.

*Quality control (QC)*. We performed standard QC steps for both array genotyped data and the subset of WGS data extracted in step 2 using PLINK[32]. Samples failing

sex check or with >3% missing data were excluded. SNPs with missing rate >3% or that failed Hardy-Weinberg Equilibrium check ($p < 1e-4$) were excluded from the study. A structure analysis was conducted to match ancestries to 1000 genomes reference haplotypes and mis-classified samples were excluded. In addition, we adopted standard TOPMed filters (https://topmed.nhlbi.nih.gov/) for variant selection. The variants that were labeled as follows were excluded: SVM (support vector machine score more negative than −0.5 and hence fails the SVM filter), CEN (falls in a centromeric region with inferred reference sequence), DISC (more than 5 percent Mendelian inconsistencies), EXHET (has excessive heterozygosity with HWE $p$-value < 1e-6) or CHRXHET (has excessive heterozygosity in male chrX).

*Combining array and WGS data.* **GAWMerge**, a protocol for integrating array and WGS data is shown in Fig. 1 and described in more detail in the **Results**. The WGS data were first prepared by extracting the selected control samples and the variants available within the array genotyping data. Utilizing the intersection of variants was important, as many false positives were introduced without this step[13]. This extraction of samples and variants was performed by BCFtools[33]. After QC, the intersection of SNPs between the array and WGS data was extracted, and the datasets were phased independently using SHAPEIT2[34,35]. The datasets were then merged using BCFtools[33].

*Imputation strategy.* The merged array and WGS data were first imputed using Minimac4[19] using the thousand genomes phase 3 version 5 EUR and AFR super populations for EA and AA samples, respectively. The reference panel includes 503 EUR and 661 AFR samples with data on GRCh37 genome version. TOPMed WGS data was converted from genome version GRCh38 to GRCh37 to match the reference and array-genotyped data. Besides applying the standard imputation quality measurement $R^2$, we also observed poorly imputed variants indicated by Empirical $R^2$ (ER²). ER² was defined only for genotyped variants as the squared correlation between leave-one-out imputed dosages and the true, observed genotypes. Under our first test for controlling type-I error (Fig. 2b), array data from COPDGene EA1 ($N = 3251$) and WGS data from ECLIPSE EA ($N = 1461$), we expected no genome-wide significant associations since all individuals were smokers and no disease was being tested between the datasets. Without the ER² filter, we found many false positives (Supplementary Fig. 1a) based around the variant on chromosome 10 (chr10:32370743, ER² = 0.391, MAF = 0.068). We recommend removing such genotyped SNPs with ER² < 0.9 from the analysis and re-running imputation without these variants included. With this and other low-quality variants removed, false positives were controlled (Supplementary Fig. 1b). With the ER² filter of 0.9, we found that 81.1% of SNPs met this criterion (Supplementary Fig. 1c) and these removed SNPs were scattered across the genome (Supplementary Fig. 1d).

*Filtering association test results.* Association analysis was conducted using rvTest[36] with ten principal components included to account for population substructure. Besides the common filters for minor allele frequency (MAF > 0.01) and imputation quality ($R^2 > 0.8$), we also investigated the imputation quality difference between array-genotyped samples and WGS-genotyped samples by comparing the imputation quality within each sample type, $R^2_{array}$ and $R^2_{WGS}$. We verified that the imputation quality between the two types of data were similar. However, some outliers ($|R^2_{array} - R^2_{WGS}| \geq 0.1$) were a major source of false positives, and were removed from the results as a post-association testing filter. Using the same test between COGEND and COPDGene EA sample comparison, inflation of GWAS $P$-values was apparent when $|R^2_{array} - R^2_{WGS}| \geq 0.1$, but otherwise no inflation was observed (Supplementary Fig. 2a). An imputation quality difference of ≥0.1 only filtered out about 5% of variants (Supplementary Fig. 2b), and the removed variants were scattered throughout the genome (Supplementary Fig. 2c).

**GAWMerge implementation.** GAWMerge was developed within the DNANexus computing environment (https://www.dnanexus.com/) and the BioData Catalyst ecosystem[37]. The protocol within the DNANexus computing environment used docker images, which have been packaged together into DNANexus applications. The BioData Catalyst ecosystem[37] protocol was implemented in the common workflow language (CWL); therefore, it is interoperable in other computing ecosystems. Both implemented workflows are built using the same docker images of the underlying software programs (https://github.com/RTIInternational/biocloud_docker_tools and https://hub.docker.com/u/rtibiocloud). The protocol has been written to easily adapt to plink or vcf formats of the genotype files, therefore either are acceptable. The BioData Catalyst workflow leverages key services, tools, and workflows available within the ecosystem including BioData Catalyst Powered by Gen3, BioData Catalyst Powered by PIC-SURE, and BioData Catalyst Powered by Seven Bridges. These tools make discovery of data for use as public controls easy with their easy-to-use web interface.

To discover optimal controls to combine with available cases, TOPMed phenotypic data were easily accessible using the Gen3 and PIC-SURE tools within the BioData Catalyst ecosystem. With these tools, users identify which studies were comparable for use as public controls, urge the access request for these studies within dbGaP, and then use as public controls with the protocol.

Computation of GAWMerge is comparable to other GWAS efforts. For example, in the analysis comparing ECLIPSE WGS data and COPDGene EA array data, phasing the 10,302 variants on chromosome 10 (overlapped with the array data) of the 1,461 samples in ECLIPSE WGS data took ~9 h using a machine with 32GB memory and 16 CPUs. The following imputation ran on a machine with 16GB memory and 4 CPUs for 2 h and 37 min. Then the re-imputation runs for similar amount of time.

**Reporting summary.** Further information on research design is available in the Nature Research Reporting Summary linked to this article.

## Data availability

The individual-level genotype and phenotype data used are all available through dbGaP. The dbGap study accession number for COGEND is phs000404, for COPDGene are phs000179 (parent study with array genotype data) and phs000951 (WGS data generated by TOPMed), and for ECLIPSE are phs001252 (parent study with array genotype data) and phs001472 (WGS data generated by TOPMed).

## Code availability

The codes to run the protocol can be found at https://github.com/RTIInternational/GAWMerge. https://doi.org/10.5281/zenodo.6841389.

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

## Acknowledgements

Primary support for developing GAWMerge, conducting analyses, and preparing the manuscript was provided by National Institute on Drug Abuse grants to E.O.J.: R01 DA044014 (PI: E.O.J.); R01 DA043980 (M-PIs: Scaheri, E.O.J., Akbarian); R01 DA051908 (M-PIs: E.O.J. and Jacobson). Support for this work was provided by the National Institutes of Health, National Heart, Lung, and Blood Institute, through the BioData Catalyst program (award 1OT3HL142479-01, 1OT3HL142478-01, 1OT3HL142481-01, 1OT3HL142480-01, 1OT3HL147154-01). Any opinions expressed in this document are those of the author(s) and do not necessarily reflect the views of NHLBI, individual BioData Catalyst team members, or affiliated organizations and institutions. Molecular data for the Trans-Omics in Precision Medicine (TOPMed) program was supported by the National Heart, Lung and Blood Institute (NHLBI). Genome sequencing for NHLBI TOPMed: COPDGene (phs000179.v6.p2) was performed at NWGC (3R01HL089856-08S1, HHSN268201600032I, and HHSN268201600032I), and Broad Genomics (HHSN268201500014C and HHSN268201500014C). Genome sequencing for NHLBI TOPMed: ECLIPSE (phs001252.v1.p1) was performed at MDI (HHSN268201600037I). Core support including centralized genomic read mapping and genotype calling, along with variant quality metrics and filtering were provided by the TOPMed Informatics Research Center (3R01HL-117626-02S1; contract HHSN268201800002I). Core support including phenotype harmonization, data management, sample-identity QC, and general program coordination were provided by the TOPMed Data Coordinating Center (R01HL-120393; U01HL-120393; contract HHSN268201800001I). We gratefully acknowledge the studies and participants who provided biological samples and data for TOPMed. The COPDGene project described was supported by Award Number U01 HL089897 and Award Number U01 HL089856 from the National Heart, Lung, and Blood Institute. The content is solely the responsibility of the authors and does not necessarily represent the official views of the National Heart, Lung, and Blood Institute or the National Institutes of Health. The COPDGene project is also supported by the COPD Foundation through contributions made to an Industry Advisory Board that has included AstraZeneca, Bayer Pharmaceuticals, Boehringer-Ingelheim, Genentech, GlaxoSmithKline, Novartis, Pfizer, and Sunovion. A full listing of COPDGene investigators can be found at: http://www.copdgene.org/directory. The ECLIPSE study (NCT00292552) was sponsored by GlaxoSmithKline. The ECLIPSE investigators included: Bulgaria: Y. Ivanov, Pleven; K. Kostov, Sofia. Canada: J. Bourbeau, Montreal; M. Fitzgerald, Vancouver, BC; P. Hernandez, Halifax, NS; K. Killian, Hamilton, ON; R. Levy, Vancouver, BC; F. Maltais, Montreal; D. O'Donnell, Kingston, ON. Czech Republic: J. Krepelka, Prague. Denmark: J. Vestbo, Hvidovre. The Netherlands: E. Wouters, Horn-Maastricht. New Zealand: D. Quinn, Wellington. Norway: P. Bakke, Bergen. Slovenia: M. Kosnik, Golnik. Spain: A. Agusti, J. Sauleda, P. de Mallorca. Ukraine: Y. Feschenko, V. Gavrisyuk, L. Yashina, Kiev; N. Monogarova, Donetsk. United Kingdom: P. Calverley, Liverpool; D. Lomas, Cambridge; W. MacNee, Edinburgh; D. Singh, Manchester; J. Wedzicha, London. United States: A. Anzueto, San Antonio, TX; S. Braman, Providence, RI; R. Casaburi, Torrance CA; B. Celli, Boston; G. Giessel, Richmond, VA; M. Gotfried, Phoenix, AZ; G. Greenwald, Rancho Mirage, CA; N. Hanania, Houston; D. Mahler, Lebanon, NH; B. Make, Denver; S. Rennard, Omaha, NE; C. Rochester, New Haven, CT; P. Scanlon, Rochester, MN; D. Schuller, Omaha, NE; F. Sciurba, Pittsburgh; A. Sharafkhaneh, Houston; T. Siler, St. Charles, MO; E.K.S., Boston; A. Wanner, Miami; R. Wise, Baltimore; R. ZuWallack, Hartford, CT. ECLIPSE Steering Committee: H. Coxson (Canada), C. Crim (GlaxoSmithKline, USA), L. Edwards (GlaxoSmithKline, USA), D. Lomas (UK), W. MacNee (UK), E.K.S. (USA), R. Tal-Singer (Co-chair, GlaxoSmithKline, USA), J. Vestbo (Co-chair, Denmark), J. Yates (GlaxoSmithKline, USA). ECLIPSE Scientific Committee: A. Agusti (Spain), P. Calverley (UK), B. Celli (USA), C. Crim (GlaxoSmithKline, USA), B. Miller (GlaxoSmithKline, USA), W. MacNee (Chair, UK), S. Rennard (USA), R. Tal-Singer (GlaxoSmithKline, USA), E. Wouters (The Netherlands), J. Yates (GlaxoSmithKline, USA). COGEND was supported by grants from the National Cancer Institute (NCI; grant number P01 CA089392, PI: L.J.B.) and NIDA (R01 DA036583 and R01 DA025888, PI: L.J.B.), both of the National Institutes of Health (NIH). Genotype data are available via dbGaP as part of the "Genetic Architecture of Smoking and Smoking Cessation" (accession number phs000404.v1.p1) and "Study of Addiction: Genetics and Environment (SAGE)" (accession number phs000092.v1.p1). Funding support for genotyping, which was performed at CIDR, was provided by 1 × 01 HG005274-01 and by the NIH Genes, Environment and Health Initiative [GEI] (U01 HG004422). CIDR is fully funded through a federal contract from the NIH to The Johns Hopkins University, contract number HHSN268200782096C. Assistance with genotype cleaning, as well as with general study coordination, was provided by the GENEVA Coordinating Center (U01 HG004446).

## Author contributions

R.M. and F.F. co-led the development of GAWMerge, all analyses, and the writing of the manuscript. NG, DBH, GPP, and EOJ conceptualized and helped develop GAWMerge. M.H.C., J.E.H., L.J.B., S.M.L., K.Y., and E.K.S. provided valuable cohort and TOPMed datasets expertize used in the development of GAWMerge. All authors edited the manuscript.

## Competing interests

E.K.S. has received institutional grant support from GlaxoSmithKline and Bayer. M.H.C. has received grant support from GSK and Bayer, and consulting or speaking fees from Illumina, Genentech, and AstraZeneca. All other authors have no competing interests.

## Additional information

## NHLBI Trans-Omics for Precision Medicine (TOPMed) Consortium

Michael H. Cho[2,3], John E. Hokanson[4], Albert V. Smith[7,8] & Edwin K. Silverman[2,3]

A full list of members and their affiliations appears in the Supplementary Information.

