## [Peer Review File · Communications Biology]

Reviewers' comments:

Reviewer #1 (Remarks to the Author):

I have read the paper by Mathur et al. The authors develop a protocol, Genotyping Array-WGS Merge (GAWMerge), to combine genotypes from arrays and whole genome sequencing for a unified analysis. The protocol is able to utilize genetic studies to increase sample size and enhance discovery from the understudied traits and ancestries.

The paper is well-written. I don't find major problems.

Reviewer #2 (Remarks to the Author):

This manuscript presents a novel protocol for combining genotypes of SNPs called with array techniques and genotypes called using whole genome sequencing (WGS). This protocol enables the use of case-only cohorts (where no control individuals have been genotyped), since publicly available WGS data can then be used as control. The paper discusses potential pitfalls by combining array and WGS data, and experiments are performed to show that the protocol works.

The idea behind the protocol is good, and the protocol seems to be relevant to the field (assuming that cohorts without controls are common, I personally don't know if this is case). The manuscript is well written, and the experiments seem to justify the protocol's correctness and usefulness. I have only have a few comments listed below.

1) The usefulness on the protocol relies on the fact that there are GWAS studies without control samples. I was curious myself as to why there exists GWAS datasets without controls? Do the authors know why? This could maybe be mentioned in the introduction to further motivate the protocol for using WGS as control.

2) I assume there could be a potential problem of combining genotypes called using arrays and WGS if some variants are called with consistently higher accuracy using array technology as compared to WGS. I'm not sure if such variants are common in the genome, but I guess it could happen if a variant can be well represented by unique (on the genome) probes on the array (these are fairly short sequences), but is positioned in a very variation-rich region, meaning that mapping e.g. 150 bp reads from WGS might be difficult. You also observe yourself that genotypes from array and WGS are not consistent. It would be nice to know why (e.g. if there are specific problematic variants, these could be filtered out). Can any experiments be performed to investigate what might cause variants to be called differently between the technologies, or is there maybe some literature on this already? Are you (given the experiments you have performed) confident that such differences can not lead to false positive associations?

3) I think it is a very good idea to start out with only the variants that are common between the array experiment and WGS, and not try to include variants from the union of the data sets.

4) You link to code for running the protocol, but it is unclear how to run or reproduce any of the experiments presented in this paper, or how to run the protocol on my own data. I would prefer that all code for reproducing the experiments is available together with clear instructions.

Reviewer #3 (Remarks to the Author):

The new GAWMerge method described by Mathur & colleagues in this manuscript is relevant to the fastly evolving GWAS data analysis field.

GAWMerge presents a feasible and sustainable solution to the problem of underrepresentation of non-European samples in GWAS data - which is mainly due to the lack of power of such results because of the small number of study participants and control sample from the general population. The integration of array and WGS genotype data to conduct GWAS with a case-only and public control design could indeed represent a feasible solution to this problem.

The method is described in detail with a clear workflow and the code is freely accessible through GitHub.

Some positive and negative aspects of the method are also critically analyzed in this work.

Please find below three minor points of critique/feedback:

The two smoking behavior studies used for the GAWMerge analysis presented in the paper, COPDGene and ECLIPSE, are both parts of TOPMed. I wonder whether the authors could elaborate a bit more on:

a) why they have chosen this phenotype over others (e.g. is this just linked to the possibility to control bias on results -ref. page 1 lines 50-51 & page 191 line 227 onwards).

b) whether they have also tested the protocol on different studies and observed a different/comparable outcome.

Would GAWMerge be compatible with targeted array association studies (disease-specific) such as those that use MetaboChip, ImmunoChip and OncoArray (amongst others)?

Page 17 lines 362-363: The authors report that the GAWMerge protocol has been written to easily adapt to PLINK or vcf formats of the genotype files. Are authors also considering interoperability with the NHGRI-EBI GWAS Catalog data-sharing standard format? Currently, the GWAS Catalog is the biggest and most frequently updated resource for disease/trait associations, with users analysing its data for an impressive list of downstream analyses.

Below we describe our point-by-point response to each issue raised and have incorporated the indicated changes in the highlighted manuscript.

Response to critiques:

Reviewer #1:

No items to update.

Reviewer #2:

Critique: 1) The usefulness on the protocol relies on the fact that there are GWAS studies without controls samples. I was curious myself as to why there exists GWAS datasets without controls? Do the authors know why? This could maybe be mentioned in the introduction to further motivate the protocol for using WGS as control.

Response: We thank the reviewer for the great suggestion. We expanded the introduction (lines 50-54) to explain the existence of case-only studies with the following: 'Case-only GWAS datasets may exist for several reasons but primary among them is that the initial study focused on phenotypes within a patient population (e.g., set point viral load among those living with HIV^{1,2} or methadone dosing among those with opioid use disorder (OUD)^{3,4}) but these case-only datasets could now be useful for GWAS of the primary disease (HIV or OUD) if paired with public controls.' Furthermore, on lines 67-68 we provided details on the number of case-only studies available in the dbGaP repositories.

Critique: 2) I assume there could be a potential problem of combining genotypes called using arrays and WGS if some variants are called with consistently higher accuracy using array technology as compared to WGS. I'm not sure if such variants are common in the genome, but I guess it could happen if a variant can be well represented by unique (on the genome) probes on the array (these are fairly short sequences), but is positioned in a very variation-rich region, meaning that mapping e.g. 150 bp reads from WGS might be difficult. You also observe yourself that genotypes from array and WGS are not consistent. It would be nice to know why (e.g. if there are specific problematic variants, these could be filtered out). Can any experiments be performed to investigate what might cause variants to be called differently between the technologies, or is there maybe some literature on this already? Are you

(given the experiments you have performed) confident that such differences can not lead to false positive associations?

Response: Thank you very much for pointing out the difference between the genotype calls from the array and WGS data. We did observe a difference by comparing the array genotyped data and WGS data using the COPDGene dataset, which has both array- and WGS-genotyped data for the same set of samples (Supplementary Figure 3). The discordance between the array genotyping platform and WGS is low, as discussed and quantified⁵. To clarify, we expanded the introduction with reference to prior literature, as follows (lines 74-76): “Although the array genotyping may have a lower overall precision due to the poor cluster separation in the genotype assignment pipeline based on a 2-dimensional metrics, the average discordant calls were below 1%⁵, which supports the feasibility to combine the array genotyping data with WGS data.”

Our design of the GAWMerge approach was informed by our previous study⁶ that devised a strategy to minimize false positive associations when combining data genotyped on different arrays. Based on our previous study, **WE** began with the intersection of variants between array and WGS data (lines 327-330). Taking the example from COPDGene European Ancestry data, the array data included 648,530 SNPs. Among them, 630,947 (97.3%) SNPs were called from the WGS data. QC steps were performed separately on the array and WGS datasets (Supplementary Table 3). Together with the post-association filtering (minor allele frequency (MAF) > 0.01; $R^2 > 0.8$; and $|R^2_{array} - R^2_{WGS}| \geq 0.1$), Supplementary Figure 4 shows that we successfully controlled the false positive rate comparing the array- and WGS-genotyped data in the COPDGene dataset.

Critique: 3) I think it is a very good idea to start out with only the variants that are common between the array experiment and WGS, and not try to include variants from the union of the data sets.

Response: We thank the reviewer for this confirmation.

Critique: 4) You link to code for running the protocol, but it is unclear how to run or reproduce any of the experiments presented in this paper, or how to run the protocol on my own data. I would prefer that all code for reproducing the experiments is available together with clear instructions.

Response: We thank the reviewer for the opportunity to clarify our protocol’s documentation. The GitHub including all the code for the protocol (<https://github.com/RTIInternational/GAWMerge>) has been further documented to describe how users can use and run the protocol.

Reviewer #3:

Critique: The two smoking behavior studies used for the GAWMerge analysis presented in the paper, COPDGene and ECLIPSE, are both parts of TOPMed. I wonder whether the authors could elaborate a bit more on: a) why they have chosen this phenotype over others (e.g. is this just linked to the possibility to control bias on results -ref. page 1 lines 50-51 & page 191 line 227 onwards). b) whether they have also tested the protocol on different studies and observed a different/comparable outcome.

Response: We thank the reviewers for pointing this out. We added our rationale for choosing COPD on lines 93-96 with the following: “COPD has well-established GWAS hits, therefore easily testing replication of signal, and it has high sample size for both European-ancestry and African-ancestry groups within the TOPMed program.”

We plan to expand the application of the GAWMerge method to other studies, like opioid use in the near future (lines 237-239).

Critique: Would GAWMerge be compatible with targeted array association studies (disease-specific) such as those that use MetaboChip, ImmunoChip and OncoArray (amongst others)?

Response: We thank the reviewer for this suggestion. The best part of using WGS data as public control is the high overlap with different types of arrays. We investigated the intersection between the TOPMed freeze.6a WGS data and the targeted arrays (Supplementary Table 2). The intersection rates between the WGS data and various custom arrays are high enough to ensure useful applicability of GAWMerge. We have added a sentence to mention the intersection rates in the main text at lines 120-123. For your convenience, we also copied Supplementary Table 2 here.

Supplementary Table 2: Overlap of TOPMed WGS data with the targeted arrays.

Array type	#SNPs on the array	#SNPs overlapped with WGS data	Overlap%
MetaboChip array	196,725	188,086	95.61%
ImmunoChip array	253,702	246,668	97.23%
OncoArray	499,170	484,288	97.02%

Critique: Page 17 lines 362-363: The authors report that the GAWMerge protocol has been written to easily adapt to PLINK or vcf formats of the genotype files. Are authors also considering interoperability with the NHGRI-EBI GWAS Catalog data-sharing standard format? Currently, the GWAS Catalog is the biggest and most frequently updated resource for disease/trait associations, with users analyzing its data for an impressive list of downstream analyses.

Response: This is a great point. The GAWMerge protocol currently requires the raw genotype data to match public control WGS data and run all the QC, phasing, imputation, and association tests. Instructions for using BioData Catalyst to identify WGS data to integrate with array data is available within the protocol’s GitHub. We have adapted GWAS Catalog data-sharing standard format as our output format to let users easily submit the results to GWAS Catalog or compare them with existing results.

References:

1. van Manen, D. *et al.* Genome-wide association scan in HIV-1-infected individuals identifying variants influencing disease course. *PLoS One* **6**, e22208 (2011).
2. Xie, W. *et al.* Genome-Wide Analyses Reveal Gene Influence on HIV Disease Progression and HIV-1C Acquisition in Southern Africa. *AIDS Res Hum Retroviruses* **33**, 597-609 (2017).
3. Lake, S. *et al.* The Cannabis-Dependent Relationship Between Methadone Treatment Dose and Illicit Opioid Use in a Community-Based Cohort of People Who Use Drugs. *Cannabis Cannabinoid Res* (2021).
4. Lo, A. *et al.* Factors associated with methadone maintenance therapy discontinuation among people who inject drugs. *J Subst Abuse Treat* **94**, 41-46 (2018).

5. Danilov, K.A., Nikogosov, D.A., Musienko, S.V. & Baranova, A.V. A comparison of BeadChip and WGS genotyping outputs using partial validation by sanger sequencing. *BMC Genomics* **21**, 528 (2020).
6. Johnson, E.O. *et al.* Imputation across genotyping arrays for genome-wide association studies: assessment of bias and a correction strategy. *Hum Genet* **132**, 509-22 (2013).

REVIEWERS' COMMENTS:

Reviewer #2 (Remarks to the Author):

All my comments and concerns have been satisfactorily addressed by the authors. I am happy with the manuscript in its current form.

Reviewer #3 (Remarks to the Author):

The authors properly addressed all my critiques and I am therefore happy with the revised version of the manuscript.